# Calibrating a network meta-analysis of diabetes trials of sodium glucose cotransporter 2 inhibitors, glucagon-like peptide-1 receptor analogues and dipeptidyl peptidase-4 inhibitors to a representative routine population: a systematic review protocol

Elaine Butterly [1], Lili Wei,[1] Amanda I Adler,[2] Saleh A M Almazam,[1] Khalid Alsallumi,[1] Luke A K Blackbourn,[3] Sofia Dias [4], Peter Hanlon [1], Katherine Hughes,[5] Jim Lewsey [1], Robert Lindsay,[6] Stuart McGurnaghan [3], John Petrie,[6] David Phillippo,[7] Naveed Sattar [8], Laurie A Tomlinson [9], Nicky Welton,[7] Sarah Wild,[10] David McAllister[1]

**Correspondence to**
David McAllister;
david.mcallister@glasgow.ac.uk

## ABSTRACT

**Introduction** Participants in randomised controlled trials (trials) are generally younger and healthier than many individuals encountered in clinical practice. Consequently, the applicability of trial findings is often uncertain. To address this, results from trials can be calibrated to more representative data sources. In a network meta-analysis, using a novel approach which allows the inclusion of trials whether or not individual-level participant data (IPD) is available, we will calibrate trials for three drug classes (sodium glucose cotransporter 2 (SGLT2) inhibitors, glucagon-like peptide-1 (GLP1) receptor analogues and dipeptidyl peptidase-4 (DPP4) inhibitors) to the Scottish diabetes register.

**Methods and analysis** Medline and EMBASE databases, the US clinical trials registry (clinicaltrials.gov) and the Chinese Clinical Trial Registry (chictr.org.cn) will be searched from 1 January 2002. Two independent reviewers will apply eligibility criteria to identify trials for inclusion. Included trials will be phase 3 or 4 trials of SGLT2 inhibitors, GLP1 receptor analogues or DPP4 inhibitors, with placebo or active comparators, in participants with type 2 diabetes, with at least one of glycaemic control, change in body weight or major adverse cardiovascular event as outcomes. Unregistered trials will be excluded.

We have identified a target population from the population-based Scottish diabetes register. The chosen cohort comprises people in Scotland with type 2 diabetes who either (1) require further treatment due to poor glycaemic control where any of the three drug classes may be suitable, or (2) who have adequate glycaemic control but are already on one of the three drug classes of interest or insulin.

## STRENGTHS AND LIMITATIONS OF THIS STUDY

⇒ Where many previous reviews have focused on fewer drugs or outcomes, the criteria used in this systematic review are designed to provide a definitive collection of phase 3 and 4 clinical trials of newer glucose lowering drugs.

⇒ The planned calibration methodology will retain the strength of trial data (not breaking randomisation) while improving representativeness using routine healthcare data and can be used to calibrate trials to any target population of interest.

⇒ Unlike other approaches to calibration, the planned approach allows more studies to be included in the analysis due to the inclusion of both individual-level participant data and aggregate-level trials potentially reducing bias.

⇒ Calibration modelling requires important assumptions, although fewer assumptions than simple extrapolation of trial results to wider populations.

⇒ Calibration could produce misleading results if applied to populations who are entirely excluded from clinical trials, not just under-represented, or in the presence of additional modifiers of treatment effect not included in the model.

**Ethics and dissemination** Ethical approval for IPD use was obtained from the University of Glasgow MVLS College Ethics Committee (Project: 200160070). The Scottish diabetes register has approval from the Scottish A Research Ethics Committee (11/AL/0225) and operates with Public Benefit and Privacy Panel for Health and Social Care approval (1617-0147).
**PROSPERO registration number** CRD42020184174.

## INTRODUCTION

Randomised controlled trials (hereafter abbreviated to trials) are the gold standard for obtaining unbiased estimates of treatment effects. However, trials are limited in terms of representativeness. Trial participants are on average younger, fitter and have fewer comorbid diseases than patients in routine care identified through disease registers, who are the target population for an intervention.[1–5]

In type 2 diabetes, one of the the most common chronic diseases,[6] this gap in representativeness is particularly evident. For example, compared with the population-based Scottish diabetes register, trial participants are on average younger and women are under-represented.[1] This under-representation remains true for trials of the newer glucose-lowering agents (sodium glucose cotransporter 2 (SGLT2) inhibitors, glucagon-like peptide-1 (GLP1) receptor analogues and dipeptidyl peptidase-4 (DPP4) inhibitors).[1 7–11] This problem of under-representation is acknowledged in clinical guidelines, for example the most recent diabetes guidance from the National Institute for Health and Care Excellence (NICE) states 'much of the evidence base used to inform this guideline has been generated from studies involving younger adults (study mean ages ranged from 45 to 68 years)'.[6]

However, the clinical implications of this under-representation are not self-evident. Since diabetes complications are more common in older age groups,[12–14] people in routine care settings may benefit more from treatments than trial participants. Alternatively, since the risk of non-cardiovascular non-diabetes-related deaths also increase with age, acting as a competing risk, routine care patients may benefit less than trial participants.[14] Therefore, in routine care settings, the applicability of trial results is uncertain.

Calibrating trial results to make them more representative of target populations in clinical practice is a promising approach to help address this uncertainty. First described by Cole and Stuart,[2] calibration involves reanalysing trial data using the prevalence of baseline characteristics in trial and target populations. Briefly, participants who, compared with the target population, are over-represented in the trial (eg, younger people) contribute less to the calibrated treatment effect estimate, while participants who are underrepresented (eg, older people) contribute more. At the expense of wider confidence intervals (CIs) for calibrated effects, this 'moves' trials in the direction of increased representativeness. Most approaches to calibration respect randomisation and so avoid the confounding by indication which can occur when estimating treatment effects using observational data which is representative of the target population. Calibration also involves fewer assumptions than simpler approaches to extrapolating trial results to target populations.

Despite these advantages, calibration has not been widely used. Until recently, calibration required individual-level participant data (IPD) (or stratification of results for all levels and every combination of baseline characteristics) for all relevant trials, making it unfeasible in most settings. We propose to overcome this problem by using a novel calibration methodology which incorporates trials where IPD are available and trials where only published summary data are available in a single model. We will use this method to compare SGLT2 inhibitors, GLP1 receptor analogues and DPP4 inhibitors in type 2 diabetes.

### Aim

To compare the efficacy of SGLT2 inhibitors, GLP1 receptor analogues and DPP4 inhibitors on glycated haemoglobin (HbA1c), body weight and cardiovascular outcomes in people with established type 2 diabetes by applying a network meta-analysis (NMA) of all relevant type 2 diabetes trials and calibrating to a selected target population from the Scottish diabetes register.

### Objectives

1. To compare the efficacy of each drug class as an add on to metformin (dual therapy)
2. To compare the efficacy of each drug class as an add on to metformin plus one other glucose-lowering drug (triple therapy)
3. To compare the efficacy of each drug class singly (monotherapy)

As well as the NMA calibrated to the routine care population, to quantify the impact of the calibration on the final results, we will also repeat 1–3 using an uncalibrated NMA and a NMA calibrated to the average population of the trials.

## METHODS

We plan to conduct a systematic review and calibrated NMA combining results from a model fitted to randomised controlled trials of the relevant drug classes (both IPD and aggregate-level data) with data for a target population defined using the Scottish diabetes register (figure 1). The start and planned end dates of the study are as follows: 29 November 2019 to 1 November 2022. Here we describe the planned systematic review, planned modelling and the characterisation of the target population.

### Systematic review
#### Eligibility criteria
This review will be performed in keeping with the Preferred Reporting Items for Systematic reviews and Meta-Analyses statement (PRISMA) statement.[15] We used the PRISMA-P checklist for this protocol.[16]

#### Population
Eligible trials will study people over 18 years old with established type 2 diabetes, with trials of healthy volunteers, people with prediabetes or trials that include any other forms of diabetes for example, type 1 diabetes or gestational diabetes, excluded. There will be no limit

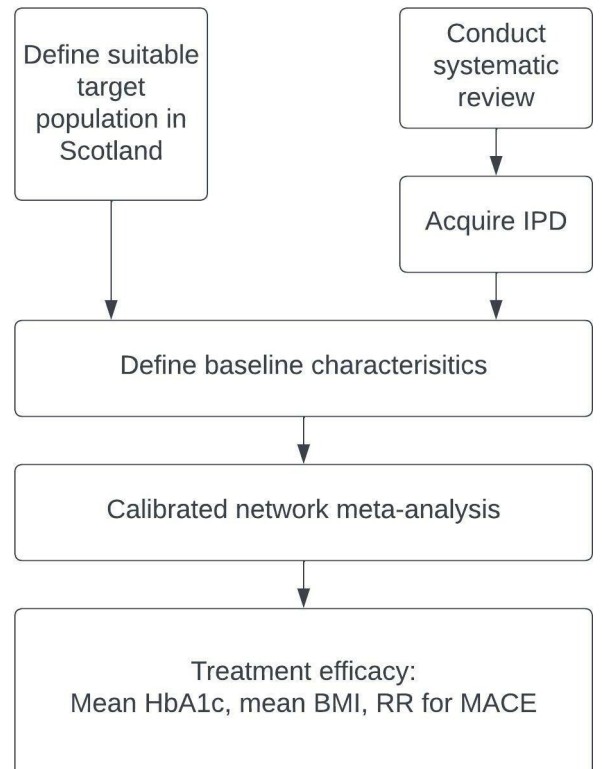

**Figure 1** Overview of output process. BMI, Body Mass Index; HbA1c, glycated haemoglobin; IPD, individual-level participant data; MACE, composite measure of major adverse cardiovascular events; RR, relative risk measures.

placed on duration or severity of diabetes and trials will be included if they examine a subpopulation of people with type 2 diabetes defined by a comorbid condition for example, a trial in people with type 2 diabetes and comorbid fatty liver disease.

## Interventions
Eligible trials will study any SGLT2 inhibitor, GLP1 receptor analogue or DPP4 inhibitor as the intervention drug. This will not be limited to drugs approved by regulatory authorities in any specific country. Intervention drug preparations can be short acting or modified release and can be prescribed as mono-therapy, dual-therapy or triple therapy with other glucose lowering drugs including the other two classes of interest plus metformin, sulfonylureas, thiazolidinediones and insulin. Trials will be excluded when the intervention drug was given as a single dose only for example, perioperative trials of GLP1 receptor analogues. Trials will be excluded if they were performed under fasting conditions for example, Ramadan specific trials, as this is likely to have influence on the treatment effect.

## Comparators
Trials will be included where the intervention drug was compared with placebo or to an active pharmacological comparator for example, metformin, sulfonylureas, thiazolidinediones, alpha-glucosidase inhibitors, SGLT2 inhibitors, GLP1 receptor analogues, DPP4 inhibitors

or insulins. Trials will be excluded if they had surgical comparators (eg, bariatric surgery) or were compared with specific non-pharmacological lifestyle interventions (eg, very low calorie diets).

## Outcomes
The trial outcomes to be analysed are:
1. Glycaemic control is measured by change in glycated haemoglobin (HbA1c) in either % or mmol/mol.
2. Change in body weight is measured by weight in kilograms or change in Body Mass index (BMI).
3. Cardiovascular outcome is measured as composite outcome such as Major Adverse Cardiovascular Events (MACE).

Trials will be included if at least one of the above outcomes was measured. It is not required to be the primary outcome of the trial. Non- inferiority trials will be included.

## Study design
Eligible trials will be limited to randomised phase 3 or 4 trials. Trial registration is required as a marker of trial quality, but no specific registration platform has to be used. There will be no limitation based on trial blinding or enrolment size. Trials of any follow-up length will be included except in cases where only HbA1c is reported, in which case these trials will only be included if the follow-up length is ≥12 weeks from randomisation reflecting the physiological turnover of red blood cells. Exploratory substudies within a trial population for example, where a small proportion of participants had an additional alternative intervention or exploratory outcome analysed, will be excluded. Other study designs including non-randomised and observational, along with existing meta-analyses, will be excluded.

## Information sources
Relevant trials will be identified by systematic searches of Medline and EMBASE (via OVID) databases using a combination of Medical Subject Headings (MeSH)[17] and keyword searches. Terms will be piloted and refined then adapted to each database (see online supplemental appendix for full search strategy).

All searches will be limited from 1 January 2002, as pilot work showed the first phase 3 trials of relevant newer glucose-lowering drugs were all commenced after this date.

To reduce the risk of publication bias, two clinical trials registries (the US clinical trials registry at https://clinicaltrials.gov/[18] and the Chinese Clinical Trial Registry at https://www.chictr.org.cn/[19]) will also be searched for eligible trials using the same criteria.

## Data management
The initial review stages will use Covidence online software[20] to manage the search records and the screening process. Eligible papers will be saved locally in pdf format and linked to the relevant trial via the corresponding trial registration identifier for example, nctid. Data extracted

from publicly available documents will be processed locally. Where IPD is available, this will be processed on Vivli Center for Global Clinical Research Data[21] or Yale University Open Data Access Project (YODA)[22] secure platforms and only approved aggregate-level results will be exported and stored on csv files locally. At the time of publication, aggregate-level data from the target population and trials will be made available along with sufficient metadata for analysis.

## Selection process

Titles and abstracts obtained from the search strategies will be screened by one reviewer for potential relevance. Where the paper is potentially relevant (or if there is uncertainty), a full-text paper will be acquired and reviewed in the next stage. A random sample of 100 titles and abstracts will be reviewed by an independent reviewer as a quality check. Two reviewers will then both independently review all full-text papers and apply eligibility criteria. Reasons for excluding papers will be documented. Where there are conflicts, papers will be discussed in a meeting with at least two reviewers and a joint decision will be documented. Where required, further information can be sought from sources such as trial registries to clarify if a paper contains a relevant trial. If a conflict of opinion remains, a third independent reviewer will be asked to review the paper. In the event an agreement cannot be made, the paper will be discussed with the steering committee for a final decision on inclusion. Papers will be included if they can be linked to a registered clinical trial meeting the eligibility criteria. The US or Chinese clinical trial registries will also be searched for relevant trials. The online registries will be searched by filtering trial condition as type 2 diabetes mellitus and trial product as any of the three classes of interest. The resulting trials will be screened for relevance using the same criteria used in the database searches. Included trials from the Medline and EMBASE searches, together with additional trials identified on the clinical trial registries, will be collated, and thereafter be identified by a unique trial registration identifier (eg, a national clinical trial id (nctid) from clinicaltrials.gov).

## Data collection process

Descriptive information for each trial (eg, intervention class, follow-up and enrolment) will be extracted from publicly available sources including online trial registries, published papers and study documents. Extraction of trial results will depend on the level of data accessible. The planned calibration analysis incorporates IPD and published trial-level aggregate data, depending on data availability. Two IPD repositories (Vivli and YODA) will be searched for data availability. Where IPD are not available, trial-level aggregate data will be collected from publicly available sources. Where the trial is registered to the US Clinical Trial registry at ClinicalTrials.gov, this can be done semiautomatically by interrogation of the Aggregate Analysis of ClinicalTrials.gov (AACT) database.[23] Results

extracted from AACT, and any code used, will be checked for accuracy by a second researcher. Where trial results are not available on AACT, they will be double extracted from published documents manually by two researchers.

## Data items

The data items to be extracted for each eligible trial are listed in table 1. This will be individual level for the IPD trials and aggregate-level data for the non-IPD trials. In trials where there is crossover or longer term follow-up with escape treatment, data will be extracted for the initial randomised period only to reduce confounding by introduction of other agents. Baseline characteristics and outcomes will also be extracted for subgroup populations in trials reporting MACE where these are available.

Table 1 Data to be collected from routine data and trial data

## Effect measures

For the IPD trials, participant-level data will be reanalysed, and effect measures calculated. Outcome data from the intention to treat data sets will be extracted where available. For non-IPD trials, all available published measures will be extracted including arm-level data (eg, HbA1c at baseline and endpoint or change in HbA1c in each arm) and treatment effect estimates (eg, analysis of covariance). For cardiovascular outcome trials, count data or proportions for each arm will be extracted for the MACE outcome.

## Risk of bias

Each eligible trial will be assessed for risk of bias using the Revised Cochrane risk-of-bias tool for randomised trials (RoB2)[24] using two independent reviewers to assess risk of bias and provide detailed information on confidence of results. Where there are conflicts, papers will be discussed in a meeting with at least two reviewers and a joint decision will be documented. Where required further information can be sought from sources such as trial registries to clarify if a paper contains a relevant trial. If a conflict of opinion remains, a third independent reviewer will be asked to review the paper.

## Calibrated NMA

We will perform calibrated NMA for the effect of treatment on glycaemic control, weight loss and cardiovascular outcome of glucose-lowering drugs from the three chosen classes. The formal details of the modelling—which is done within a Bayesian framework—are explained in detail in a separate publication,[25] and the use of the modelling for this particular application is described in the supplementary appendix. Briefly, the modelling is conducted in two stages. First, a model is fitted to the trial IPD and trial aggregate-level data to estimate treatment effects adjusting for differences in effect modifiers in the trial populations. Next, the fitted model is applied to the target population data to estimate treatment efficacy in that population.

**Table 1** Data variables for extraction

| Individual-level data/aggregate-level data | Trial descriptive data |
|---|---|
| **Baseline characteristics/target population characteristics**<br>▲ Age in years<br>▲ Sex<br>▲ Race/ethnicity<br>▲ Geographical location<br>▲ Duration of type 2 diabetes mellitus in years .<br>▲ Known cardiovascular disease—does the participant have history of atherosclerotic cardiovascular disease (coronary, cerebrovascular or peripheral vascular disease—see ICD10 codes below*) (y/n) .<br>▲ Use of any concomitant glucose lowering agents at baseline (y/n)<br>▲ Use of metformin at baseline (y/n)<br>▲ Use of sulphonylurea at baseline (y/n)<br>▲ Use of thiazolidinediones at baseline (y/n)<br>▲ Use of DPP4 inhibitor at baseline (y/n) .<br>▲ Use of GLP1 receptor agonist at baseline (y/n)<br>▲ Use of SGLT2 inhibitor at baseline (y/n)<br>▲ Use of insulin at baseline (y/n)<br>▲ Smoking status—current smoker (y/n) eGFR (document what calculation used if this info is available)<br>▲ Serum creatinine (in case eGFR needs to be calculated)<br>▲ Urinary albumin creatinine ratio in mg/g<br>▲ Total/HDL/LDL cholesterol in mmol/l<br>▲ Systolic blood pressure (mm Hg)<br>▲ Diastolic blood pressure (mm Hg)<br>▲ Body weight (kg)<br>▲ Height (m)<br><br>**Trial outcomes**<br>▲ HbA1c (% or mmol/mol)<br>▲ Body Mass Index (kg/m$^2$)<br>▲ Body weight and height (kg, m)<br>▲ Cardiovascular outcome: MACE composite endpoint where available along with individual components (usually cardiovascular death, non-fatal myocardial infarction and non-fatal stroke—but check study documents) and hospitalisation for heart failure where available<br>▲ Adverse events | **Trial details**<br>▲ Trial identifier<br>▲ Trial title<br>▲ Brief description<br>▲ Phase<br>▲ Single vs multicentre trial<br>▲ Geographical location of trial<br>▲ Study design (eg, blinded)<br>▲ Sponsor (eg, Industry sponsor)<br>▲ Date of trial completion<br><br>**Eligibility**<br>▲ Inclusion/exclusion criteria<br>▲ Study enrolment number<br><br>**Intervention**<br>▲ Intervention drug (generic names)<br>▲ Comparator drug(s)/regimes<br>▲ Dosage and frequency of medications<br>▲ Other drugs allowed in arms<br>▲ Duration of follow-up for each outcome independently |

Note: All variables will be extracted in available units (which will be recorded) then can subsequently converted onto the desired scale.
*ICD10 codes[27] to identify cardiovascular disease: ICD10: I20.0, I20.1, I20.8, I20.9, I21.0, I21.1, I21.2, I21.3, I21.4, I21.9, I22.0, I22.1, I22.8, I22.9, I23.0, I23.1, I23.2, I23.3, I23.4, I23.5, I23.6, I23.8, I24.0 I24.1, I24.8, I24.9, I25.0, I25.1, I25.2, I25.3, I25.4, I25.5, I25.6, I25.8, I25.9, I63.0, I63.1, I63.2, I63.3, I63.4, I63.5, I63.6, I63.8, I63.9, I64.0, I65.0, I65.1, I65.2, I65.3, I65.8, I65.9, I66.0, I66.1, I66.2, I66.3, I66.4, I66.8, I66.9, I67.2, I67.8, I67.9, I69.3, I69.4, I69.8, I70.0, I70.1, I70.2, I70.8, I70.9, I73.0, I73.1, I73.8, I73.9.
DPP4, dipeptidyl peptidase-4; eGFR, estimated glomerular filtration rate; GLP1, glucagon-like peptide-1 ; HbA1c, glycated haemoglobin; HDL, high-density lipoprotein; ICD, International Classification of Diseases ; LDL, low-density lipoprotein; MACE, major adverse cardiovascular events; SGLT2, sodium glucose cotransporter 2.

For the main analysis, we will include the following covariates: age, duration of diabetes, HbA1c, estimated glomerular filtration rate (eGFR), total cholesterol, high-density lipoprotein cholesterol, systolic blood pressure, diastolic blood pressure, BMI, sex, ethnicity, smoking status, history of cardiovascular disease, history of heart failure, metformin use and insulin use. For the routine data target population, we will use multiple imputation in order to account for missingness. We anticipate low missingness in the IPD, therefore we will conduct a complete case analysis for the trial data. In sensitivity analyses, we will examine the robustness of the findings to alternative covariate choices.

In the model fitting stage, there is a single underlying individual-level model across all studies, which includes terms for treatment effects, prognostic covariate effects and treatment–covariate interactions (subgroup effects), with intercepts stratified by study terms for treatment effects and subgroup effects (t–c interactions). This is fitted directly to individuals in the IPD studies, and for the aggregate-level trials the individual-level model is integrated over the trial covariate distribution to obtain the aggregate-level model. Such integration requires information on the distribution of the covariates in each trial (ie, the distribution of age, sex, BMI, etc). Summary-level information on the marginal covariate distributions is normally available in the table of baseline characteristics of published trial manuscripts; and can be extracted as part of the systematic review and, together with estimated covariances from the IPD studies used to obtain the joint distributions of covariates for each trial.[25 26]

A trial with IPD contributes more information about treatment–covariate interactions than a trial with only aggregate-level data. Nonetheless, both IPD and aggregate-level trials contribute information to the estimation of such interactions. More importantly, the modelling approach enforces consistency for trials with both types of data, and assumes that the relative treatment effects are, conditional on the known covariate levels, similar across trials. This assumption is weaker than the standard assumption in NMA that relative treatment effects are *unconditionally* similar across trials.

If we assume that treatment–covariate interactions are equal (or exchangeable) for treatments within drug-classes, we can estimate these treatment–covariate interactions if there are one or more trials with IPD (or multiple aggregate-level trials across a range of covariate values) within each drug class. This assumption is also weaker than the standard assumption in NMA that there are no covariate–treatment interactions (at least for covariates not identically distributed in all trials).

In the second stage of calibration, the model results are applied to the target population data to estimate treatment efficacy in that population. This is achieved by integrating the estimated regression model for covariate-specific treatment effects over the joint covariate distribution in the target population to obtain population-specific treatment effects. This can be done for any target population where information is available for the relevant covariates (or even for any hypothetical target population where an investigator is interested in the treatment effects for any given set of covariate levels).

In this calibrated NMA, for the MACE outcome only, we propose to extend the modelling by including subgroup-level effect estimates for those aggregate trials where subgroup-level data are available for the main covariates of interest. This will further improve precision and potentially allow for the equal interactions within classes assumption to be relaxed. The existing model framework allows inclusion of single subgroups (or fully factorial subgroups where effects are presented for all combinations of the covariates of interest). We will extend the modelling to the more usual scenario where potentially correlated subgroup effects are presented for multiple subgroups.

## Target population

A key step in calibration is defining a specific target population of interest. For this research, the Scottish diabetes register has been used to identify the target population. The Scottish diabetes register (SCI Diabetes) includes data from >99% of people in Scotland with a diagnosis of diabetes. The data in the Scottish diabetes register platform are extracted regularly from a national database that collates information relevant to diabetes from primary and secondary care that is linked to other data sets via deterministic linkage. Linked data sets include hospitalisations and deaths, prescribing and dispensing data and national renal and cancer registries providing a rich and representative data set. For this calibration, the 2019 extract from SCI Diabetes was used. The target population comprises people in Scotland with type 2 diabetes who either (1) require further treatment due to inadequate glycaemic control (HbA1c≥53 mmol/mol) where any of the three drug classes may be suitable (ie, no contraindications of any of the three classes and no alternative focus of treatment for example, end of life), or (2) have adequate glycaemic control but are already on one of the three drug classes of interest or insulin. People with type 2 diabetes in Scotland alive on 1 January 2019 with diabetes duration of at least 1 year were first identified (n=256 620). The cohort was then refined by applying eligibility criteria in a stepwise fashion (Section 2, online supplemental appendix). The criteria were agreed with the steering committee and will be finalised prior to any extraction of results from the clinical trial data. For some of the criteria, for example, for BMI and renal function, several cut-off options were examined to assess their impact on the final population characteristics, and the final decision chosen based on clinical judgement. For the clinical contraindications, diseases were identified via a combination of International Classification of Disease (ICD) codes[27] from hospital admission data, prescribing data and outpatient clinic attendance data. Additional details are provided in the online supplemental appendix.

Subsequently, we will calibrate the trial results to specific subpopulations within the overall target population to estimate subpopulation treatment effects. We will identify these subpopulations using clustering methods alongside clinical judgement and public engagement.

We described the impact of each step in the selection process on the population characteristics using means and SD, medians and IQR and counts and percentages for symmetrically distributed continuous variables, skewed continuous variables and binary variables respectively.

### Patient and public involvement

We have involved diabetes patient and public involvement (PPI) groups in the design and funding application stages of this work. The groups provided positive feedback and improved the readability of plain English summary of the study. We will continue to involve a local PPI group to help guide the subpopulations for calibration and to ensure any outputs are distributed in the best way to help people with diabetes. We have also invited people with diabetes to join our steering committee.

### RESULTS
### Target population

We present the details of the target population chosen for calibration to ensure this is documented prior to any data analysis from the clinical trials. Initially, the whole population of people with type 2 diabetes in Scotland alive as of 1 January 2019 with a duration of diabetes of at least 1 year was identified (n=256 620). This cohort was 56.2% male with a mean age of 66.7 (SD, 12.7) years and had been diagnosed with type 2 diabetes for a mean (SD) of 10.0 (7.1) years. Mean (SD) BMI was 31.7 (6.6) kg/m$^2$, 14.5% were current smokers and 21.2% had a previous history of cardiovascular disease.

This whole population was then restricted to those defined as eligible for treatment escalation and in whom any of the three classes would be considered, to define the final target population for calibration (n=127 992). This group was 60.6% male with a mean age of 63.8 (12.1) years and had been diagnosed with type 2 diabetes for a mean of 10.3 (6.9) years. Mean (SD) BMI was 32.7 (6.2) kg/m$^2$, 14.6% were current smokers and 17.6% had a previous history of cardiovascular disease (table 2).

### DISCUSSION

This study will use all the available IPD and aggregate-level trial data and data from a diabetes register to estimate the effectiveness of SGLT2 inhibitor, GLP1 receptor analogue and DPP4 inhibitors for patients in routine care settings. It will also be the first study, to the best of our knowledge, to calibrate multiple trials (both IPD and aggregate level) to a representative target population defined using routinely collected healthcare data.

We have opted to perform calibrated NMA to address the differences in characteristics of people with type 2 diabetes in UK clinical practice and participants in trials for the included drug classes.[1 7–11] A previous study, on applying trial eligibility for a major cardiovascular outcome trial of an SGLT2 inhibitor to a primary care database (n=1 238 909), found that only 15.7% of people with type 2 diabetes had similar levels of cardiovascular risk to participants in the trial, and only 11.1% of SGLT2 inhibitor treated patients were comparable to the trial participants in terms of baseline characteristics.[9] Similarly, a European study (n=803 836) of German, Norwegian, Swedish and Dutch populations found that the proportion who were eligible for the four main SGLT2 inhibitor cardiovascular outcome trials ranged from 17% to 59%.[10] Similarly, in the USA, a cross-sectional study evaluating the eligibility of people within the Diabetes Collaborative Registry (n=172 643) for SGLT2 inhibitor cardiovascular outcome trials reported that 48% of their population were ineligible for any one of the trials, with individual trial eligibility ranging from 26% to 44%.[8] However, while such studies quantified the degree of lack of representativeness, they do not allow us to assess the likely impact of such differences to the underrepresented populations.

An existing approach to address lack of representativeness is to estimate treatment effects using observational data. Known as pharmaco-epidemiology, such analyses commonly use routine healthcare data, where included individuals are more representative by definition. However, while such studies have mostly yielded results similar to those from comparable clinical trials,[28] they have led to inaccurate conclusions in several cases; this is because pharmaco-epidemiological analyses are limited by the problem of confounding by indication; treated and untreated patients differ in their susceptibility to disease-related outcomes and it is currently not possible to determine when such analyses have successfully overcome confounding by indication.[29–32]

Another existing approach commonly used in health technology assessments is to use simple extrapolation, wherein relative treatment effects from clinical trials are combined with data on event rates from other (ideally representative) data sources to estimate benefits and harms.[33] For example, on applying a relative risk of death of 0.80 from a clinical trial to a target population with 1 year mortality of 10%, the expected absolute risk reduction for that target population is 2%. A strength of using simple extrapolation is that doing so avoids the problem of confounding by indication. However, in this approach, one must assume that treatment effects are similar in trial and routine care populations regardless of differences in patient characteristics. This assumption can rarely be justified on biological or clinical grounds.

Trial calibration has the advantage over pharmacoepidemiology that it avoids confounding by indication (as it does not break randomisation). However, it also has an advantage over simple extrapolation that it does not require the assumption that treatment effects are similar in trial and routine care settings, but only that treatment effects are similar for participants who are similar with

**Table 2** Summary characteristics of the Scottish target population

| | Whole Scottish type 2 diabetes population ≥1 year postdiagnosis | Defined Scottish target population for calibration |
|---|---|---|
| n | 256 620 | 127 992 |
| n (%) male | 144 338 (56.2%) | 77 599 (60.6%) |
| Mean (SD) age in years | 66.7 (12.7) | 63.8 (12.1) |
| Mean (SD) Body Mass Index in kg/m$^2$ | 31.7 (6.6) | 32.7 (6.2) |
| Mean (SD) duration diabetes in years | 10.0 (7.1) | 10.3 (6.9) |
| Mean (SD) HbA1c in mmol/mol; % | 60.2 (15.1). 7.7% (1.7%) | 67.4 (13.1). 8.3% (1.2%) |
| Mean (SD) estimated glomerular filtration rate in mL/min/1.73 m$^2$ | 77.6 (21.3) | 81.7 (19.9) |
| Mean (SD) total cholesterol in mmol/L | 4.3 (1.0) | 4.4 (1.0) |
| Mean (SD) high density lipoprotein cholesterol in mmol/L | 1.2 (0.3) | 1.1 (0.3) |
| Mean (SD) systolic blood pressure in mm Hg | 135.5 (12.2) | 135.7 (11.9) |
| Mean (SD) diastolic blood pressure in mm Hg | 76.8 (7.6) | 77.8 (7.6) |
| White racial group (%) | 75.8 | 75.6 |
| Asian racial group (%) | 3.8 | 4.2 |
| Black racial group (%) | 0.5 | 0.5 |
| Mixed or other racial group (%) | 3.0 | 3.1 |
| Race unknown (%) | 17.0 | 16.5 |
| Current smokers (%) | 14.5 | 14.6 |
| History of heart failure (%) | 11.9 | 9.1 |
| History of cardiovascular disease (%) | 21.2 | 17.6 |
| Metformin use (%) | 55.2 | 69.2 |
| Insulin use (%) | 8.4 | 10.3 |

Summary characteristics of the whole population within the Scottish diabetes register who have been diagnosed with type 2 diabetes for at least 1 year and a subset who are the defined target population. Missingness of these individual variables in the target population are as follows: sex (0%), age (0%), BMI (3.4%), duration (0%), HbA1c (1.2%), estimated glomerular filtration rate (1.5%), total cholesterol (2.1%), high-density lipoprotein cholesterol (7.4%), systolic blood pressure (1.1%), diastolic blood pressure (1.1%) and smoking status (7.4%).

HbA1c, glycated haemoglobin.

respect to characteristics included in the modelling (eg, with similar age, sex, BMI, etc). This assumption allows greater confidence in applying trial data to routine care settings suggesting that, alongside other methodologies (including pharmaco-epidemiology), it may have a valuable role in assessing the likely applicability of trial findings to participants in routine care settings.

The particular form of trial calibration we propose to use—calibrated NMA via multilevel network metaregression—was recently developed to address limitations of conventional NMA (combining IPD and aggregate-level data in a coherent manner, while exploring and explaining heterogeneity in treatment effects according to differences in participant characteristics within and across trials) by coauthors (DP, NW and SD). We are not aware that this approach has previously been used to address the under-representation within clinical trials of participants with certain characteristics. This approach to calibrated NMA has a number of advantages. First, it allows us to calibrate trial findings to any target population, including subpopulations of clinical interest,

provided one is willing to assume that treatment effects are similar for participants who are similar with respect to characteristics included in the modelling. Indeed, it even allows us to calibrate treatment effects to notional populations, allowing us to explore the sensitivity of trial findings under any plausible set of patient characteristics. Perhaps more importantly, the method allows us to use all relevant trials, whether or not IPD are available. This is likely to result in greater precision and less bias than methods which require access to IPD for all calibrated trials.

### Limitations
Calibrated NMA is a potentially useful approach for assessing the applicability of trial findings to target populations in routine care settings, however, in addition to the assumptions stated above, there are a number of limitations. First, if target populations and trial participants differ, even after conditioning on known characteristics, with respect to other characteristics which modify treatment effects, the calibrated

treatment effects may be biased. Second, even where all important characteristics are included in the modelling, if they are incorrectly parameterised this may also cause misleading findings (eg, if the true treatment–covariate interaction is non-linear and it is modelled with a linear term). Finally, some participants are entirely excluded from clinical trials, not simply under-represented (eg, those with very severe frailty, extensive multimorbidity, or overwhelming personal circumstances such as severe dependence syndromes). While it may be technically possible to calibrate trial results to these groups (eg, by extrapolation), findings from such analyses are likely to be misleading.

## Ethics and dissemination

Ethical approval for use of individual participant-level data was obtained from the University of Glasgow MVLS College Ethics Committee (Project No: 200160070). The Scottish diabetes register has approval from the Scottish A Research Ethics Committee (ref 11/AL/0225) and operates with Public Benefit and Privacy Panel for Health and Social Care approval (ref 1617-0147).

**Author affiliations**
[1]School of Health and Wellbeing, University of Glasgow, Glasgow, UK
[2]Diabetes Trials Unit, University of Oxford, Oxford, UK
[3]Institute of Genetics and Molecular Medicine, University of Edinburgh, Edinburgh, UK
[4]Centre for Reviews and Dissemination, University of York, York, UK
[5]Department of Diabetes, Glasgow Royal Infirmary, NHS Greater Glasgow and Clyde, Glasgow, UK
[6]University of Glasgow BHF Glasgow Cardiovascular Research Centre, Glasgow, UK
[7]Population Health Sciences, University of Bristol, Bristol, UK
[8]Institute of Cardiovascular and Medical Sciences, University of Glasgow, Glasgow, UK
[9]Epidemiology and Population Health, London School of Hygiene and Tropical Medicine, London, UK
[10]Public Health Sciences, University of Edinburgh, Edinburgh, UK

**Acknowledgements** We would like to acknowledge the people with diabetes in Scotland whose data make the diabetes register and the healthcare professionals who record these data while providing healthcare. We would also like to thank the University of Glasgow Medical and Veterinary Life Sciences patient and public involvement and engagement group for their input regarding this study. We would finally like to acknowledge and thank Daniel Kavanagh, MBChB student at the University of Glasgow, for his involvement in the early screening stages of this review.

**Contributors** DM, DP, SD and NW designed the study and EB and LW designed the search strategy. EB and LW will conduct the systematic review screening. EB, LW, KA and SAMA will conduct the risk of bias scoring. LAKB and SM provided cleaned data and supported data acquisition in the Scottish diabetes register. EB conducted the analysis within the Scottish diabetes register, supervised by DM. EB and DM wrote the first draft of this protocol paper. AIA, PH, KH, JL, RL, JP, NS, LAT and SW are all expert members of the projects steering committee who have contributed to planning, design decisions requiring clinical input and results interpretation from the Scottish diabetes register. AIA, LAKB, PH, KH, JL, RL, SM, JP, NS, LAT, SW, LW, SAMA, KA, SD, DP and NW reviewed, commented on and made changes on the final protocol and manuscript.

**Funding** This research is funded by the Medical Research Council (MR/T017112/1: Routine care treatment effectiveness in people with type 2 diabetes: maximising the applicability of clinical trials). David McAllister is funded via an Intermediate Clinical Fellowship and Beit Fellowship from the Wellcome Trust. None of the funders had any influence over the study design, analysis or decision to submit for publication.

**Competing interests** DMcA, EB, DP, SD, NW report funding for the present manuscript from the Medical Research Council (MRC) (MR/T017112/1). DMcA reports funding from the Wellcome Trust. PH reports funding from the MRC. AA reports departmental support for a randomized controlled trial from NovoNordisk to look at a GLP-agonist in a disease area other than diabetes. LB reports investment funds with shares held in various companies including Novo Nordisk and Astrazeneca, however these are chosen at the discretion of the fund manager and will fluctuate over time. No stocks/funds were chosen to benefit specifically from any company with a relationship to this manuscript. JP reports institutional grants via Merck KGaA and Novo Nordisk (co-fund with JDRF), institutional supply of investigational medicinal product for clinical trials from AstraZeneca, and personal fees from Boehringer Ingelheim (via IQVIA and via ACI Clinical), Biocon, Novo Nordisk and Merck. JP is also a member of an advisory board for Novo Nordisk and has an unpaid leadership role for Novo Nordisk UK Research Foundation (former Chair of Board of Trustees). NS reports institutional grants from AstraZeneca, Boehringer Ingelheim, Novartis and Roche Diagnostics and personal fees from Abbott Laboratories, Afimmune, Amgen, AstraZeneca, Boehringer Ingelheim, Eli Lilly, Hanmi Pharmaceuticals, Janssen, Merck Sharp & Dohme, Novartis, Novo Nordisk, Pfizer, and Sanofi. LT reports MRC, Wellcome and NIHR funding, funding from GSK for an epidemiological study of kidney disease (no personal payment received) and reports consulting for Bayer in relation to an observational study of chronic kidney disease (no personal payment received). LT is also a member of the MHRA Expert advisory group (Women's Health) and a member of 3 non-industry funded (NIHR/MRC) trial advisory committees (unpaid). NW reports honoraria for training and masterclasses on statistical methodology from Association of British Pharmaceutical Industries, Campbell Ireland, Centre for Global Development, NICE International and NICE Scientific Advice. All other authors report no competing interests.

**Patient and public involvement** Patients and/or the public were involved in the design, or conduct, or reporting or dissemination plans of this research. Refer to the Methods section for further details.

**Patient consent for publication** Not required.

**Provenance and peer review** Not commissioned; externally peer reviewed.

**Data availability statement** Data may be obtained from a third party and are not publicly available. The data required for this study will come from multiple sources. Published and publicly available trial data will be available from journal publications, clinical trial registries and published study documents. The IPD used will be available through application to third-party repositories including Vivli Center for Global Clinical Research Data and Yale University Open Data Access Project (YODA). The data for target population calibration from the Scottish diabetes register are not publicly available. At the time of publication, aggregate-level data from the target population and trials will be made available along with sufficient metadata for analysis.

**ORCID iDs**
Elaine Butterly http://orcid.org/0000-0001-9410-0237
Sofia Dias http://orcid.org/0000-0002-2172-0221
Peter Hanlon http://orcid.org/0000-0002-5828-3934
Jim Lewsey http://orcid.org/0000-0002-3811-8165
Stuart McGurnaghan http://orcid.org/0000-0002-3292-4633
Naveed Sattar http://orcid.org/0000-0002-1604-2593
Laurie A Tomlinson http://orcid.org/0000-0001-8848-9493

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
