## [Reviewer comments · BMJ Open]

ARTICLE DETAILS

TITLE (PROVISIONAL)	Calibrating a network meta-analysis of diabetes trials of sodium glucose co-transporter 2 inhibitors, glucagon-like peptide-1 receptor analogues and dipeptidyl peptidase-4 inhibitors to a representative routine population: a systematic review protocol
AUTHORS	Butterly, Elaine; Wei, Lili; Adler, Amanda; Almazam, Saleh; Alsallumi, Khalid; Blackburn, Luke; Dias, Sofia; Hanlon, Peter; Hughes, Katherine; Lewsey, Jim; Lindsay, Robert; McGurnaghan, Stuart; Petrie, John; Phillippo, David; Sattar, Naveed; Tomlinson, Laurie; Welton, Nicky; Wild, Sarah; McAllister, David

VERSION 1 – REVIEW

REVIEWER	Neri Alvarez-Villalobos Universidad Autonoma de Nuevo Leon, Subdireccion de Investigación
REVIEW RETURNED	09-Aug-2022

GENERAL COMMENTS	This protocol lays the groundwork for a systematic review and calibrated network meta-analysis that will assess the efficacy of glucose-lowering agents (SGLT-2 inhibitors, GLP-1, and DPP-4 inhibitors) in type 2 diabetes. Authors propose a novel approach that will allow the inclusion of trials whether the individual-level participant data is available or not. Furthermore, the authors will calibrate the retrieved trials to the Scottish diabetes register to identify the target population. The protocol does a good job explaining the overall approach, as well as stating its limitations. Nevertheless, there are some issues in the protocol: In the methods section, selection process subheading, the authors mention that two independent reviewers will apply eligibility criteria to identify the trials that will be included in the analysis. Authors should also state how the research team will address the conflicts between each author's criteria if there are any (stated in the PROSPERO registry). This can also be added in the abstract and the risk of bias subheading. Page 7, 5th line (methods heading, systematic review subheading) the authors state that eligible trials will study people over 18 years old, nevertheless the PROSPERO registration states, "but if included people <18 years can still include". Although inviting patients with diabetes to join the steering committee and involving a PPI group makes the research more patient-centered, the authors should state in detail how, either the PPI group or the steering committee, was involved during the whole research. Authors should also check that the page numbers stated in the PRISMA-P checklist match the protocol pages. Overall, the study was an interesting read and proposes a novel approach with a patient-centered goal in mind.
---

REVIEWER	Yuki Matsuhashi Hirosaki University School of Medicine Graduate School of Medicine
REVIEW RETURNED	10-Aug-2022

GENERAL COMMENTS	I think this is very interesting project to compare the efficacy and difference of glycemic control, body weight management and reduction the risk of cardiovascular disease, while the oral antidiabetic drugs (OADs: DPP-4 inhibitors, GLP-1 receptor agonists and SGLT2 inhibitors) were used in routine clinical. These OADs are often used in Japan, too. In routine clinical, these background of patients are very wide and difference, age(young ~ old), constitution(lean ~ obese), duration of diabetes, presence of diabetic and arteriosclerotic complications. I sometimes thought that these patients background might affect the efficacy of OADs - " the right person in the right place". So, I truly expect the outcomes of this trial.
---

VERSION 1 – AUTHOR RESPONSE

Reviewer 1	This protocol lays the groundwork for a systematic review and calibrated network meta-analysis that will assess the efficacy of glucose-lowering agents (SGLT-2 inhibitors, GLP-1, and DPP-4 inhibitors) in type 2 diabetes. Authors propose a novel approach that will allow the inclusion of trials whether the individual-level participant data is available or not. Furthermore, the authors will calibrate the retrieved trials to the Scottish diabetes register to identify the target population. The protocol does a good job explaining the overall approach, as well as stating its limitations. Overall, the study was an interesting read and proposes a novel approach with a patient-centered goal in mind.	Many thanks for your comments. We are pleased our study is of interest.
	In the methods section, selection process subheading, the authors mention that two independent reviewers will apply eligibility criteria to identify the trials that will be included in the analysis. Authors should also state how the research team will address the conflicts between each author's criteria if there are any (stated in the PROSPERO registry). This can also be added in the abstract and the risk of bias subheading.	We have updated the main document to include the following on page 8 and 9. “Where there are conflicts, papers will be discussed in a meeting with at least two reviewers and a joint decision will be documented. Where required further information can be sought from sources such as trial registries to clarify if a paper contains a relevant trial. If a conflict of opinion remains, a third independent reviewer will be asked to review the paper. In the event an agreement cannot be

		made, the paper will be discussed with the steering committee for a final decision on inclusion.
	Page 7, 5th line (methods heading, systematic review subheading) the authors state that eligible trials will study people over 18 years old, nevertheless the PROSPERO registration states, “but if included people <18 years can still include”.	Thank you for highlighting this point. We have updated the PROSPERO registration to reflect the decision to only include trials conducted in adults ≥ 18 years old
	Although inviting patients with diabetes to join the steering committee and involving a PPI group makes the research more patient-centered, the authors should state in detail how, either the PPI group or the steering committee, was involved during the whole research.	We have updated the main document to include the following on page 11/12. “We have involved diabetes patient and public involvement (PPI) groups in the design and funding application stages of this work. The groups provided positive feedback and improved the readability of plain English summary of the study. We will continue to involve a local PPI group to help guide the subpopulations for calibration and to ensure any outputs are distributed in the best way to help people with diabetes. We have also invited people with diabetes to join our steering committee.” More detail on the exact involvement of the PPI group and person with diabetes in the steering committee will be available with the publication of results.
	Authors should also check that the page numbers stated in the PRISMA-P checklist match the protocol pages.	The PRISMA P checklist has been updated.
Reviewer 2	I think this is very interesting project to compare the efficacy and difference of glycemic control, body weight management and reduction the risk of cardiovascular disease,	Many thanks for your comments. We are pleased the study is of interest and clinically relevant.

	while the oral antidiabetic drugs (OADs: DPP-4 inhibitors, GLP-1 receptor agonists and SGLT2 inhibitors) were used in routine clinical. These OADs are often used in Japan, too. In routine clinical, these background of patients are very wide and difference, age(young ~ old), constitution(lean ~ obese), duration of diabetes, presence of diabetic and arteriosclerotic complications. I sometimes thought that these patients background might affect the efficacy of OADs - " the right person in the right place". So, I truly expect the outcomes of this trial.	
--	---	--

VERSION 2 – REVIEW

REVIEWER	Neri Alvarez-Villalobos Universidad Autonoma de Nuevo Leon, Subdireccion de Investigación
REVIEW RETURNED	21-Sep-2022

GENERAL COMMENTS	Thank you for authors for your response on the review of your systematic review protocol "Calibrating a network meta-analysis of diabetes trials of sodium glucose co-transporter 2 inhibitors, glucagon-like peptide-1 receptor analogues and dipeptidyl peptidase-4 inhibitors to a representative routine population: a systematic review protocol". Almost all the changes were done as suggested, either to the manuscript or to the manuscript's PROSPERO entry. Authors did not change the word "commoner" to "more common". "Commoner" refers to an ordinary person, without rank or title and it's not the correct way to write the comparative form of the adjective common.
--